# N-Amidation of Nitrogen-Containing Heterocyclic Compounds: Can We Apply Enzymatic Tools?

**DOI:** 10.3390/bioengineering10020222

**Published:** 2023-02-07

**Authors:** Anran Yang, Xue Miao, Liu Yang, Chao Xu, Wei Liu, Mo Xian, Huibin Zou

**Affiliations:** 1State Key Laboratory Base of Eco-Chemical Engineering, College of Chemical Engineering, Qingdao University of Science and Technology, Qingdao 266042, China; 2CAS Key Laboratory of Bio-Based Materials, Qingdao Institute of Bioenergy and Bioprocess Technology, Chinese Academy of Sciences, Qingdao 266101, China; 3Institute of Corrosion Science and Technology, Guangzhou 510535, China

**Keywords:** enzymatic catalysis, amide bond forming, nitrogen-containing heterocycle

## Abstract

Amide bond is often seen in value-added nitrogen-containing heterocyclic compounds, which can present promising chemical, biological, and pharmaceutical significance. However, current synthesis methods in the preparation of amide-containing N-heterocyclic compounds have low specificity (large amount of by-products) and efficiency. In this study, we focused on reviewing the feasible enzymes (nitrogen acetyltransferase, carboxylic acid reductase, lipase, and cutinase) for the amidation of N-heterocyclic compounds; summarizing their advantages and weakness in the specific applications; and further predicting candidate enzymes through in silico structure-functional analysis. For future prospects, current enzymes demand further engineering and improving for practical industrial applications and more enzymatic tools need to be explored and developed for a broader range of N-heterocyclic substrates.

## 1. Introduction

Nitrogen-containing heterocycles contain at least one nitrogen in their ring framework, most of which exhibits important biological and pharmacological properties [1]. N-heterocycle structure presents diversified inter-molecular interactions, such as hydrogen bonding, dipole–dipole interaction, π-stacking interaction, hydrophobic interaction, etc.; thus, many small molecule drugs containing N-heterocycle structure to interact with targeting enzymes or receptors [2]. According to the registration information by the Food and Drug Administration (FDA), N-heterocycles drugs accounted for more than 59% of all approved drugs in 2014, and more than 80% of drugs containing one or more N-heterocyclic nucleus among the 200 best-selling small molecule drugs in 2018 [3,4].

Another advantage of N-heterocyclic structure is that it can provide an active ligation site to prepare functional derivatives, and N-amidation is quite often seen in the synthesis of N-heterocyclic drugs [5]. According to the data from the FDA, around two-thirds of registered small-molecule drugs present amide bond groups [6]. For example, N-amidation is found in the well-sold N-heterocycles drugs of Lisinopril, Ramipril, Enalapril, Ketoconazole, Loratadine, Sitagliptin, and Doxazosin (Figure 1). Aiming at high conversion rates, low levels of epimerization and limited by-products, N-amidation of heterocyclic compounds still relied on the well-developed chemical methods [7,8], such as acid-catalyzed Schmidt reaction [9,10], Ritter reaction [11], Schotten–Baumann reaction, Passerini reaction [12], Ugi reaction [13,14], and Beckmann rearrangement [15]. More green methods were also developed, which utilized more efficient catalysts [16]. Take ruthenium catalyst as an example; this catalyst can promote the coupling of amines and alcohols in the absence of any acids, bases, and stoichiometric oxidant, producing hydrogen as the only by-product [17]. However, in the preparation of structurally complex chemicals, the majority of these chemical methods have vast amounts of byproducts to present overall low atomic economy [18,19,20]. For example, the preparation of 1 kg of the anti-HIV drug Fuzeon (containing amide bond) through chemical process requires 45 kg of raw materials, excluding the solvents used in the synthesis and purification steps [17]. Thus, recent enzymatic methods with higher selectivity and efficiency were fast developed [20]. This study reviewed the current chemical and enzymatic methods for the N-amidation of nitrogen-containing heterocyclic compounds and predicted potential enzymatic tools, which can be applied in this area through in silico analysis.

## 2. Chemical Methods

### 2.1. N-Heterocycles Production by Metal Catalysts

Metal-catalyzed cross-coupling reactions are quite often utilized in the preparation of complex N-heterocycles [21]. For example, a study synthesized carbazoles from anilides and boronic acids with NNO pincer-type Pd(II) complex B catalyst under aerobic condition in aqueous solution. This green method highlights the advantage of a recyclable catalyst, wide substrate range, one-pot process, high yield, and low catalyst stacking. The minimum loading of the catalytic rate can reach 0.0001 mol%, the catalyst can still be recycled six times under this circumstance [22]. The detailed information is summarized in Figure 2A.

Another study utilized the cooperative Cu(II)/Fe(II) catalyst, which could catalyze the Ritter-type C−H activation followed by amination to produce a series of amide-containing N-heterocycles [23]. This method is applicable to a wide range of substrates that enable the rapid synthesis of amide ligation on nitrogen-containing heterocyclic compounds, including co-synthetic precursors for two anxiolytic drugs: Pagoclone and Pazinaclone. The detailed information is summarized in Figure 2B.

Enantioselective metal catalyst was also developed. For example, chiral catalysts were applied in the enantioselective addition reaction for the synthesis of hemiaminal intermediates, and the intramolecular cyclization can process under mildly basic conditions with excellent enantioselectivities and high yields [24]. The detailed information is summarized in Figure 2C.

### 2.2. Transition Metal Catalysis

Transition metal catalysts were applied in amides forming from alkenes, amines and CO, with no stoichiometric by-products. Based on the pioneering studies [25,26], a recent study established a cooperative catalytic system with transition metal catalysts [27], and the method would synthesis target products in mild conditions. The detailed information is summarized in Figure 2D.

### 2.3. Cycloaddition Methods

Cycloaddition is another popular strategy in the preparation of N-heterocycles. Cycloaddition prefers Lewis acid/base or chiral catalysts other than metal catalysts. For example, a diarylprolinol silyl ether catalyst was applied as chiral catalyst in the aza-Diels–Alder reaction to produce bicyclic aza-heterocyclic products with high enantioselectivity, [28]. However, one of the weaknesses of the aza-Diels–Alder cycloaddition is that it usually demands chiral precursors to promote the cycloaddition [29]. The detailed information is summarized in Figure 2E.

### 2.4. Photocatalytic Methods

More recent photocatalytic protocols were applied in N-amidation reaction through radical aminocarbonylation. For example, alkyl amides and amide-containing N-heterocycles could be prepared by metal photoredox catalysts, and the reaction can occur under visible light irradiation [30,31]. Non-metal organic photoredox catalysts were also developed. For example, organophotoredox catalyst eosin Y was applied in the synthesis of alkyl N-amides from thioamides [32]. The detailed information is summarized in Figure 2F,G, respectively.

## 3. Enzymatic Tools

### 3.1. Nitrogen Acetyltransferase

In the area of synthetic chemistry, alternative enzymatic tools are in fast development, with the advantages of high selectivity and enantioselectivity under mild conditions [33,34,35]. Some enzymes were found to catalyze the N-amidation reaction for N-heterocycle substrates. Nitrogen acetyltransferase (N-acetyltransferase) is one of the enzymatic tools known to catalyze the amidation reaction towards a broader range of substrates, from small molecules, such as aminoglycoside, to macromolecules of various proteins [36]. Earlier study found that one N-acetyltransferase (pyrB) can catalyze the N-acetylation on small N-heterocyclic compounds (imidazole) in which acetyl coenzyme A was utilized as acetyl donor [37]. A recent study reported that the N-acetyltransferase FbsK can catalyze the N-acetylation in the biosynthesis of fimsbactin (having N-amide heterocyclic structure) [38].

However, N-acetyltransferase has limitation in the in vitro system, as the acyl donor (acetyl coenzyme A or succinyl coenzyme A) is not an economic raw material for the in vitro production. The second limitation of N-acetyltransferase is that it can only catalyze acetylation (acetyl coenzyme A as acyl donor) on or succinylation (succinyl coenzyme A as acyl donor) during amide bond formation.

### 3.2. Carboxylic Acid Reductase

Carboxylic acid reductase (CAR) is another well-known biocatalytic tool, especially in the biosynthesis of aldehydes, alcohols, and alkanes [39]. Recent studies found that CAR can be applied in the formation of amide bonds, including the N-amidation, towards a broader range of substrates including N-heterocycles (piperidine, piperazine, homopiperazine) [40,41,42]. Different with N-acetyltransferase, which utilizes acetyl-coenzyme A in formation of amide structures, CAR utilizes carboxylic acid to start the bio-conversion. In detail, CARs consist of three parts: core-domain (A-domain), peptidyl carrier protein (PCP), and reduction domain (R-domain). In A-domain, carboxylic acid is activated by ATP to form acyl-AMP complex and then is further converted to acyl-thioester by phosphopantetheine thiol (Figure 3). If acyl-thioester is not transferred to R-domain for the biosynthesis of aldehyde (after reduction by NADH), it can react with the nucleophilic reagent, such as amine, to generate target amides. Thus, engineered CAR (only contains A-domain) is often utilized in amides formation [39,40]. For example, engineered CAR*mm*-A (from *Mycobacterium maritimum*) can be utilized in vitro to form amides (including N-heterocycles) from a broader range of substrates. As ATP is demanded to activate CAR, class III polyphosphate (CHU) is included in the in vitro system to regenerate ATP [43,44]. At optimized conditions (175 mM piperazine or homopiperazine, 1 mM trans-cinnamic acid, 3 mg/mL CAR*mm*-A lysate, 100 mM MgCl_2_, and CHU as ATP circulatory system), the yield of corresponding amides can reach 99%. When the amount of acyl donor (carboxylic acid) increased, the yield reduced to 74% [43,44].

The catalytic mechanism of CARs was explored in recent studies. In general, it is believed that acyl-AMP is important for nucleophilic attack during the catalytic process [45]; however, the exact mechanism of CARs in amide forming or lactamization needs to be further investigated.

Comparing with N-acetyltransferase, CAR has several advantages: (1) can be utilized in the in vitro system; and (2) CAR can recognize a broader range of substrates, including long chain fatty acids [46]. However, the practical studies for the amidation reaction by CAR is limited, and further research is demanded in the area [43].

### 3.3. Lipase and Cutinase

Lipase and cutinase all belong to the α/β-hydrolase superfamily, their typical catalytic function is to form or hydrolyze ester bond [33,47,48]. Lipase is widely applied in the industry due to its temperature, solvent tolerance, and broad catalytic properties, and amide bond lysis/formation is one of its untypical properties [33]. Similarly, cutinase can also be applied for the synthesis or hydrolysis of amides [40,48]. Comparing with lipase, cutinase has the same Ser-His-Asp catalytic triad [49], but does not have the hydrophobicity lip [50], thus the active serine residue may be directly exposed to environments for broader substrates.

Thermo and solvent stability is one of advantages of lipases and cutinases as amide-forming catalysts. The stability may benefit from the non-polar hydrophobic interactions between protein side chains with resistance to the breaking of hydrogen bonds in their three dimensional structures [51]. The zinc binding sites, which are also responsible for the enhanced thermal stability [52]. For example, *Bacillus* lipase shows excellent stability in hydrophobic organic solvents, even in 10–50% (*v*/*v*) of short chain alkanes, benzene and toluene [53]. *Fusarium solani pisi* cutinase can tolerate high temperature (70 °C) and organic solvent [54]. In addition, structures and mechanisms of lipase and cutinase were intensively studied [55,56] as referable factors for further modification and engineering. The commercial accessibility, broad substrate specificity, broad reaction type, and highly selective and specific absence of any cofactor dependence, could work in mild conditions [57,58,59].

Cutinase from different sources present high diversity. For example, fungal cutinase has Mw of approximately 22–26 kDa, with an optimum pH of 10 and optimum temperature of 30–40 °C. While bacteria cutinase has larger Mw (about 30 kDa), with an optimum pH of 8.5 to 10.5 and a higher thermal stability than fungal cutinase [60], it was indicated that bacterial cutinase may be preferable in amide forming application.

In the amide forming process, different with N-acetyltransferase and CAR, which demand CoA or ATP to activate the acyl donor during amide bond formation, lipase and cutinase can catalyze the amidation through the acyl-enzyme intermediate. During biocatalytic amidation by lipase or cutinase (Figure 4), acyl donor (carboxylic acid, ester, and vinyl acetate) firstly interacts with the Ser-His-Asp catalytic triad to form tetrahedral intermediate; then acyl group ligates the serine residue to form acyl-enzyme intermediate; at last, the acyl-enzyme intermediate can be attacked by nucleophile amine to form an amide bond [61,62].

In practical application, *Candida antarctica* lipase A (CAL-A) could catalyze the N- acetylation on methyl 2-piperidinecarboxylate in the persistent of PrCO_2_CH_2_CF_3_ and TBME [63]. CAL-A (Mw 45 kDa, pI 7.5) has a large catalytic pocket with the catalytic triad of Ser184-Asp334-His366, the oxyanion hole (Gly185 and Asp95), and hydrophobic lid (residues 217–308). CAL-A exhibits excellent thermal stability (up to 90 °C) and the lid may be important for the acylation behavior of CAL-A [64,65,66].

Although the N-amidation activities of lipase and cutinase were not reported for nitrogen-containing heterocyclic substrates, this study speculated a lipase/cutinase catalyzed amidation process towards piperazine (Figure 4). The testing experiments are in process by our lab and will be published later.

As the formation of acyl-enzyme intermediate is important for the amidation reaction by lipase and cutinase, this study further in silico analyzed the known lipases and cutinases by using the AutoDock tool [67] to screen candidate lipases or cutinases, which can form acyl-enzyme intermediates with the active acyl donor of vinyl acetate (Figure 5). Some positive results are summarized in Figure 5 and Table 1.

## 4. Concluding Remarks and Future Prospects

In summary, this study demonstrates the feasibility of enzymatic tools (nitrogen acetyltransferase, carboxylic acid reductase, lipase, and cutinase) for selective amide bond formation towards N-heterocyclic compounds under mild conditions, which may present an advantage towards currently known chemical methods (Table 2).

The enzymatic method provides a highly selective alternative to current synthesis methods of amide-containing N-heterocycles from variable acyl donors and amines without side reactions observed in chemical methods. The enzymatic method provides a valuable starting point, although candidate enzymes have limitations in current applications in organic synthesis: (1) nitrogen acetyltransferase and CAR demand co-enzyme or co-factor (CoA and ATP) in amidation; (2) lipase and cutinase may have side-reactions (ester-bond formation), and their amide forming activities towards N-heterocyclic substrates demand further testing.

To improve the current known enzymatic tools, the candidate enzymes demand further engineering and modification for practical applications in organic synthesis and other relevant fields [79,80]. For example, although lipase is one of the industrial enzymatic tools in organic synthesis with the advantages of stereoselectivity and resistance to extreme reaction conditions [65], the binding pocket of natural lipase demands further engineering for the target N-heterocyclic substrates for efficient amidation reaction [56]. One candidate strategy is to replace the residues of long chains with the residues of shorter chains engineering the substrate binding pocket. As seen in the engineering of lipase (PDB: 1CEX), which expands the workable substrates [81]. On the contrary, if residues with larger side chains in binding pockets are engineered, the substrate range will be decreased [82]. Considering the properties of N-heterocyclic substrate, another candidate engineering strategy is to include aromatic residues in the binding pocket, thus increasing the π–π stacking interaction with N-heterocyclic substrate [83]. Fusion strategy can also be utilized in engineering robust amidation bio-tools. For example, the ICCG cutinase was confused with α-synuclein (αSP) in its C-terminal to facilitate its adhesion to PET [84]. A similar strategy can be applied in engineering evolved cutinase to reduce steric hindrance to N-heterocyclic substrate. Other than the well-known lipase CAL-B, CAL-A is another promising starting lipase for further evolution [85,86]. For further improvement, the acyl donor-binding pocket demands further engineering to increase its acyltransferase activity [87], which is important for the amidation reaction by lipase (Figure 4).

As coenzyme or cofactor are demanded for some enzymatic tools (N-acetyltransferase and CAR) in amide forming process, robust coenzyme or cofactor regeneration systems demand to be developed and applied for these enzymatic tools. Similar with the ATP regeneration system (CHU) in the CAR catalyzing process [43,44], acyl coenzyme A regeneration system is demanded for future application of N-acetyltransferase in the N-amidation of N-heterocyclic compounds. A flexible polyphosphate-driven regeneration system may be suitable to perform this work [88]. The system has the advantage of recognizing a broader range of acyl donors to regenerate corresponding acyl coenzyme A, thus both acetyl coenzyme A and succinyl coenzyme A can be regenerated for amidation reaction catalyzed by N-acetyltransferase.

For future prospects, more amide ligating enzymes need to be screened for their potential application in amidation of a broader range of N-heterocyclic substrates. Similar with CAR, major amide ligases are ATP-dependent [62,89], they work in aqueous condition and demand ATP to form acyl-adenylate intermediates to activate amide bond formation. Amino acid ligases and non-ribosomal peptide synthetases are representatives for this group and majorities of them were approved in the biosynthesis of natural metabolites, such as nikkomycin [90], bacilysin [91], tabtoxin [92], rhizocticin [93], and shinorine [94]. Another group of amide-forming biocatalysts are ATP-independent serine proteases (including transpeptidases) and lipases [61,62,95,96], which do not demand costly cofactors and can perform a broader range of amide-forming reactions. For example, transpeptidases can incorporate exogenous building blockings into peptidoglycan via forming fresh amide ligation [95,96]. The strategy may be applicable once we develop novel enzymatic tools and design novel synthetic routes towards amide-containing N-heterocyclic compounds.

However, majorities of these amide-forming enzymes (amino acid ligases, peptide synthetases, and proteases) have a limited substrate scope, and their catalytic feasibilities towards N-heterocyclic substrates demand further research, judgment, and improvement. In view of industrial and scale-up amidation techniques, the enzymatic tools and methods must compare with corresponding chemical tools and methods [97], of their cost, efficiency, safety, and toxicity. It is generally accepted that enzymatic tools may have advantages when amidation steps are required in the in vitro production of structurally complex natural products.

In conclusion, limited enzymatic tools were reported in the preparation of amide-containing N-heterocyclic compounds. With the fast development of enzyme engineering and biotechnology, more robust enzymes will be discovered and engineered to improve their performance and application, which in turn will reduce the barrier and cost in design and produce value-added N-heterocyclic compounds.

## Figures and Tables

**Figure 1 bioengineering-10-00222-f001:**
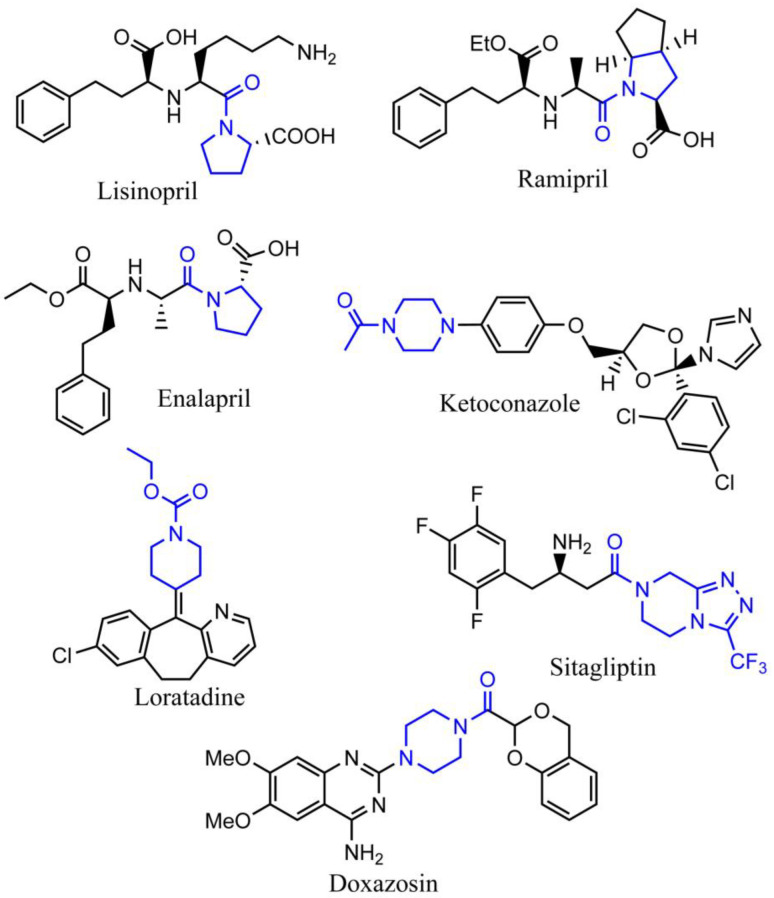
The representative marketing drugs that have amide ligation on the nitrogen-containing heterocycle structure. The blue color indicates the featured structure of each molecule.

**Figure 2 bioengineering-10-00222-f002:**
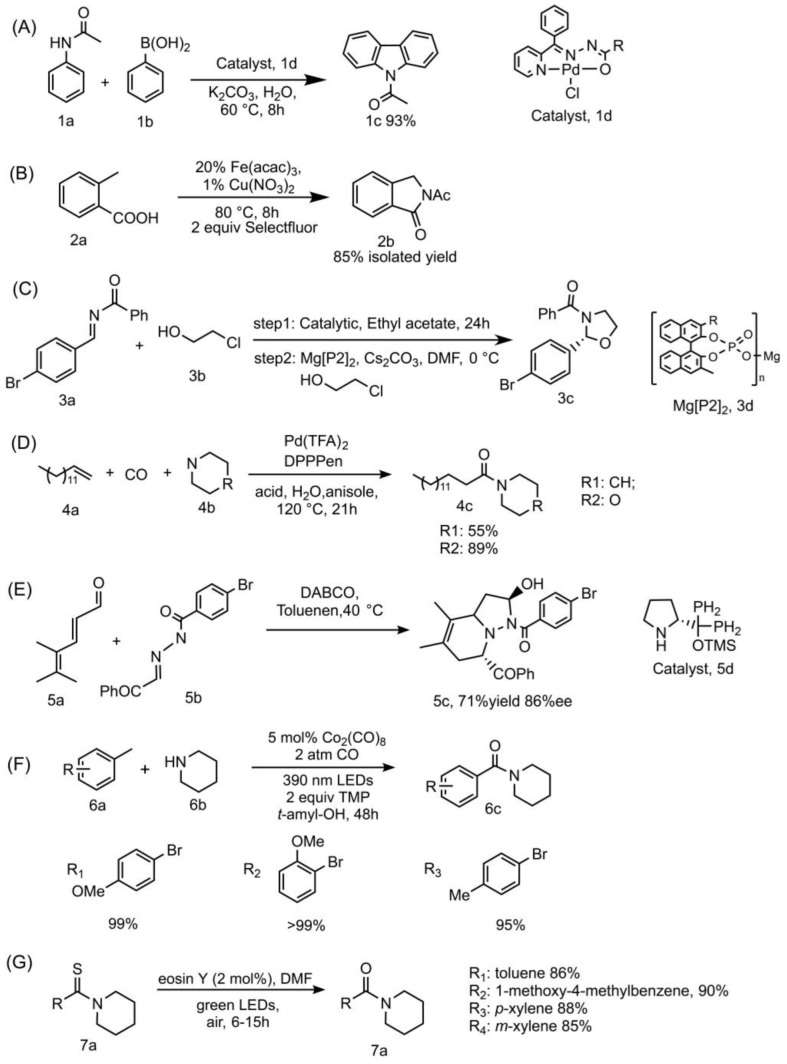
Details of representative chemical amide forming methods towards N-heterocyclic compounds. (**A**) acetanilide (1a, 2 mmol), phenylboronic acid (1b, 2 mmol), Pd(II) complexes catalyst (1d, 0.01 mol%), K_2_CO_3_ (4 mmol), solvent (water), 60 °C, 8 h, yield of the target product (1c) can reach 93 %; (**B**) 2-methylbenzoic acid (2a) (27.2 mg, 0.2 mmol, 1 equiv), selectfulor (141.7 mg, 0.4 mmol, 2 equiv), catalyst (20% Fe(acac)_3_, 1% Cu(NO_3_)_2_), Schlenk tube, 80 °C, 8 h, the yield of product (2a) can reach 85%. (**C**) 3a (1.0 equiv), 3b (2.0 equiv), catalyst (2.5 mol%), solvent (ethyl acetate). After step 1, ethyl acetate is removed; DMF and Cs_2_CO_3_ (2.0 equiv) are added. The yield of product (3c) can reach 92%, 95%ee. (**D**) 4a (0.8 mmol), CO (10 atm), 4b (0.4 mmol), Pd(TFA)_2_ (0.01 mmol), DPPPen (0.012 mmol), NH_2_CH_2_CO_2_Me·HCl (0.04 mmol), (HCHO)n (0.04 mmol), solvent (anisole, 1.0 mL), 120 °C, 21 h. (**E**) 5a (0.2 mmol), 5b (0.1 mmol), 5c (0.02 mmol), DABCO (0.02 mmol), solvent (toluene, 0.5 mL), 40 °C. (**F**) 5 mol % Co_2_(CO)_8_, 2 atm CO, 2 equiv TMP, 390 nm LEDs. (**G**) 7a (1.0 mmol), eosin Y catalyst (2 mol%), solvent (DMF, 3 mL), and green LEDs (2.6 W, 161 lm) irradiation.

**Figure 3 bioengineering-10-00222-f003:**
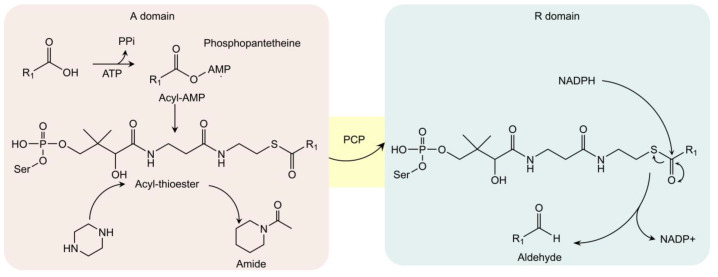
The A-domain of carboxylic acid reductase (CAR) presents amidation activity towards N-heterocycles. The A, R domains and peptidyl carrier protein (PCP) are shown in red, blue, and yellow. In the absence of NADPH, acyl-thioester intermediates react with variable amines (including piperazine) for amide formation in the A domain. On the other hand, the acyl-thioester intermediate is eventually reduced (by NADPH) into aldehyde in the R domain.

**Figure 4 bioengineering-10-00222-f004:**
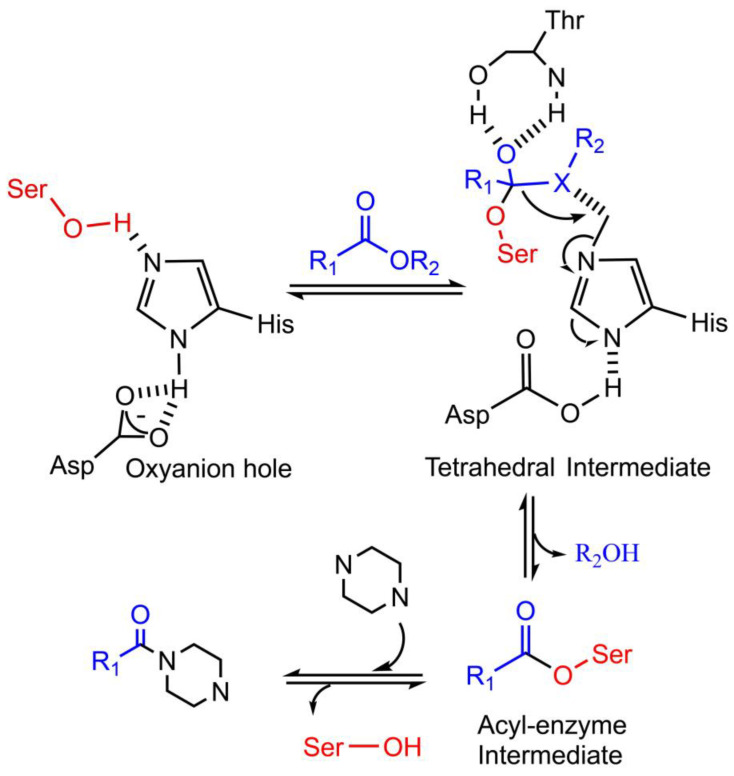
Mechanism of a speculated lipase/cutinase catalyzed amidation reaction towards piperazine. The Ser-His-Asp catalytic triad center can activate the acyl donor to form a tetrahedral intermediate, which is stabilized by the oxyanion hole. Then, this high-energy transition state product decomposes into acyl-enzyme intermediate. The acyl-enzyme intermediate is further attacked by nucleophilic piperazine to obtain N-amidated piperazine.

**Figure 5 bioengineering-10-00222-f005:**
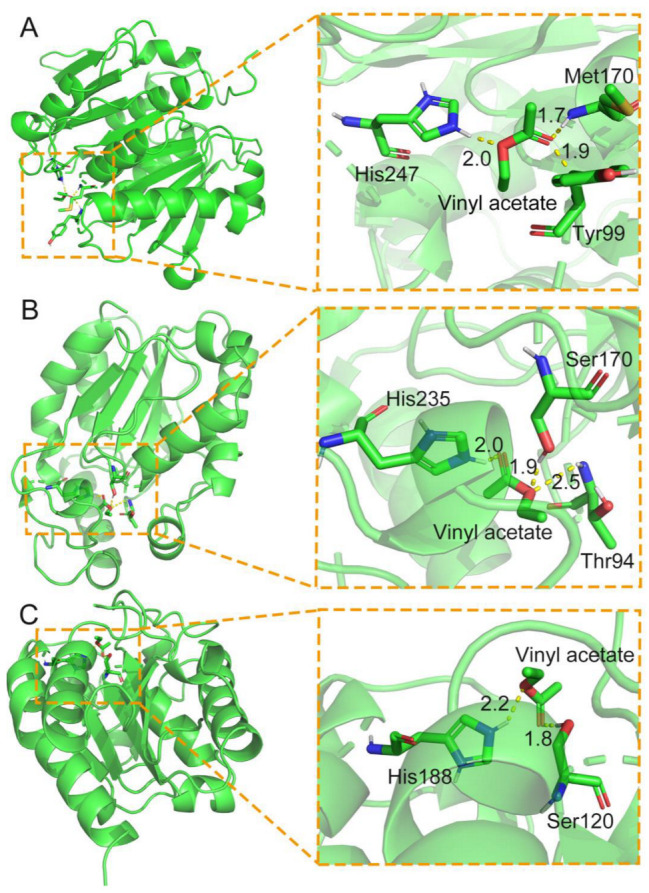
Autodock analysis of representative cutinases, which can form acyl-enzyme intermediates with the acyl donor of vinyl acetate. (**A**) Docking of vinyl acetate in the structure of cutinase (PDB: 6AID). (**B**) The primary sequence of one *Aspergillus oryzae* cutinase (NCBI: OOO09588.1) was modelled in its structure by SWISS-Model [68,69] before docking analysis with vinyl acetate. (**C**) Docking of vinyl acetate in the structure of cutinase (PDB: 1FFB).

**Table 1 bioengineering-10-00222-t001:** Candidate cutinases and lipases for the bio-amidation towards N-heterocyclic substrates.

Enzyme	Source	Description	Reference
Cutinase, PDB: 6AID	*Thermobifida alba*	Indicated by docking analysis (Figure 4a), this enzyme has potential ability to form acyl-enzyme intermediate for further amidation towards N-heterocyclic substrates.	[70]
Cutinase, NCBI: OOO09588.1	*Aspergillus oryzae*	Indicated by docking analysis (Figure 4b), this enzyme has potential ability to form acyl-enzyme intermediate for further amidation towards N-heterocyclic substrates.	Not published yet
Cutinase, PDB: 1FFB	*Fusarium solani pisi*	Indicated by docking analysis (Figure 4c), this enzyme has potential ability to form acyl-enzyme intermediate for further amidation towards N-heterocyclic substrates.	[71]
Cutinase, PDB: 4OYY and 4OYL	*Humicola insolens*	This enzyme can form acyl-enzyme intermediate with acyl donor. In addition, the aromatic ring in the catalytic pocket can help to form π–π stacking interactions with N-heterocyclic rings.	[48,72]
Cutinase, NCBI: HQ147785	*Thermobifida cellulosilytica*	The catalytic triad is located near to the surface of this enzyme, which helps to accommodate acyl donors and N-heterocyclic rings.	[73]
Cutinase, PDB: 3GBS	*Thielavia terrestris*	Both hydrophilic and hydrophobic residues are found in the catalytic pocket, which contribute to recognize a broader range of acyl donors and amine substrates (including N-heterocyclic substrates).	[74]
Cutinase, NCBI: AB445476.2	*Thermobifida alba*	In the substrate binding pocket, the arrangement of alternating hydrophobic and hydrophilic amino acids can facilitate its binding to acyl donors and hydrophilic amine substrates.	[75]
Lipase, NCBI: AEA86017	*Pseudomonas stutzeri*	The enzyme can catalyze the monoacetylation reaction towards the N-heterocyclic substrate of piperazine.	[76]
PETase, NCBI: GAP38373.1	*Ideonella sakaiensis*	This PETase has a more open and wider substrate binding pocket than cutinase and lipase. The S238F mutation can further provide π–π stacking interactions towards N-heterocyclic substrates.	[77,78]

**Table 2 bioengineering-10-00222-t002:** Comparison of chemical and enzymatic methods in amides forming of N-heterocyclic substrates.

	Advantage	Disadvantage	Reaction Condition
Chemical methods	Matured techniques, high yield and efficiency.	Require extreme conditions (high temperature, strong acid/base), and large amounts of by products and solvents.	Detailed information is summarized in Figure 2.
Nitrogen acetyltransferase	Mild condition, workable in aqueous solvent, and high selectivity.	Demands high-cost acyl coenzyme A (acetyl coenzyme A or succinyl coenzyme A) supplementation.	In aqueous solution, 0.05 M potassium phosphate buffer, pH 7.4, mild temperature, the concentrate of acyl CoA up to 5 × 10^−5^.
Carboxylic acid reductase	Mild condition, workable in aqueous solvent, and recognizes a broader range of substrates for amide bond forming.	Demands ATP circulatory system.	In aqueous solution, carboxylic acid (1–10 mM), amine (175 mM), MgCl_2_ (100 mM), CAR*mm*-A (3 mg/mL), Tris-buffer (100 mM), polyphosphate (4 mg/mL), pH 8.5, 30 °C, 16 h.
Lipase and Cutinase	Thermo and organic solvent tolerance, industrial resources, do not demand expensive supplementation such as acyl CoA and ATP, and can recognize broad range of acyl donors.	N-amidation activities demand further research for nitrogen-containing heterocyclic substrates.	In organic solvents, broad temperature range (40–70 °C), broad acyl donors (acid, esters, and vinyl acetate).

## Data Availability

Not applicable.

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
