# Peer review of "N-Amidation of Nitrogen-Containing Heterocyclic Compounds: Can We Apply Enzymatic Tools?"

_bioengineering, 2023, doi:10.3390/bioengineering10020222_

Round 1

Reviewer 1 Report

The manuscript is an interesting review on the enzymatic amidation of nitrogen heterocycles and can be accepted for publication after the following corrections:

The authors should check the numbers of the references in the text: for example Ref 40 refers to Carboxylic acid reductases, not to cutinases. In Table 1, the ref. 54 does not correspond to lipase, but to cutinase from Thermobifida alba. Ref 62 refers to cutinase and this should be mentioned.  The sentence in line 202 does not correspond to ref 64, but to ref 65.

In Line 209 “although” should be deleted

Author Response

Correspondence to Reviewer 1

Comment 1.  The authors should check the numbers of the references in the text: for example Ref 40 refers to Carboxylic acid reductases, not to cutinases. In Table 1, the ref. 54 does not correspond to lipase, but to cutinase from Thermobifida alba. Ref 62 refers to cutinase and this should be mentioned.  The sentence in line 202 does not correspond to ref 64, but to ref 65.

Response to comment 1.  Thank you for the comment. We carefully checked the citation and reference list and corrected wrong notes and citation.

Comment 2.  In Line 209 “although” should be deleted

Response to comment 2.  We deleted “although” in last paragraph.

Reviewer 2 Report

N-amidation of nitrogen-containing heterocyclic compounds: can we apply enzymatic tools?

The review article's subject is interesting because not many papers deal with this problem. And the advantages of enzymatic synthesis are undoubted stereospecificity and selectivity under mild conditions, provided that the appropriate enzyme is used. In the work submitted for review, I have the impression that the authors lacked an idea of how to present the topic.

The work definitely needs to be corrected/supplemented according to the following points:

1. In the chemical methods part, the authors gave only a few examples of the metal-catalyzed formation of amides containing the N-heterocyclic fragment. There should definitely be more examples in this chapter because nothing is mentioned about the rhodium-catalyzed, zirconium-catalyzed, gold-catalyzed, zinc-catalyzed, and cobalt-catalyzed reactions. There is also no information about metal-free conditions like quaternary-ammonium-salt-catalyzed reactions. Moreover, no reaction scheme is given in this section. I realize that this review does not apply to these methods of amide bond formation, but their contribution cannot be so diminished, even more so in the review paper.

2. The graphics in the work are prepared in different fonts and should be unified.

3. In the enzymatic tools chapter, the authors should place more examples of the applicability of chosen enzymes, not only in the field of amide bonds because for this topic it is only a few examples but in general applications.

Green Synthesis and Catalysis, Volume 3, Issue 3, August 2022, Pages 294-297; Org. Biomol. Chem., 2010, 8, 886–895; Synbio 2023, 1, 54–64.

Author Response

Reviewer 2

Comment 1.  In the chemical methods part, the authors gave only a few examples of the metal-catalyzed formation of amides containing the N-heterocyclic fragment. There should definitely be more examples in this chapter because nothing is mentioned about the rhodium-catalyzed, zirconium-catalyzed, gold-catalyzed, zinc-catalyzed, and cobalt-catalyzed reactions. There is also no information about metal-free conditions like quaternary-ammonium-salt-catalyzed reactions. Moreover, no reaction scheme is given in this section. I realize that this review does not apply to these methods of amide bond formation, but their contribution cannot be so diminished, even more so in the review paper.

Response to comment 1.  Thank you for the comment. We expanded this section in revised manuscript to present more detailed chemical methods. In support of this section, a new figure (figure 2) is also added.

Comment 2.  The graphics in the work are prepared in different fonts and should be unified.

Response to comment 2.  Thank you for the comment. We revised all the figures to unify the font and size, also modified some figures to improve their qualities.

Comment 3.  In the enzymatic tools chapter, the authors should place more examples of the applicability of chosen enzymes, not only in the field of amide bonds because for this topic it is only a few examples but in general applications.

Response to comment 3. Thank you for the comment. We added more detailed information and cases to support the enzymatic section of the revised manuscript. We also added one table (table 2) to compare different enzymatic tools.

Reviewer 3 Report

The manuscript reviews the enzyme-catalyzed N-amidation of nitrogen-containing heterocyclic compounds. They focus on reviewing the feasible enzymes (nitrogen acetyltransferase, carboxylic acid reductase, lipase, and cutinase) for the amidation of N-heterocyclic compounds. Using enzymes as biocatalysts in organic reactions is very promising and has attracted the attention of many research groups. Therefore, this review is important and will help the readers to understand the role of enzymes in the catalysis of N-amidation of nitrogen-containing heterocyclic compounds.
However, the manuscript is short and some details are missing. For example, it is necessary to describe the advantages of using enzymes in organic reactions.
- reaction conditions of chemical and enzymatic reactions should be compared (e.g., Temp., pH, solvent, time).
- many related original and review papers are not cited.
- a comprehensive table should be added that summarizes all enzyme-catalyzed N-amidation of nitrogen-containing heterocyclic reactions containing experimental conditions and yields.
Generally, I found the manuscript a chemical review rather than an enzymatic review. Enzymatic details are fragile.

Author Response

Reviewer3

Comment 1. Question: However, the manuscript is short and some details are missing. For example, it is necessary to describe the advantages of using enzymes in organic reactions.

Response to comment 1.  Thank you for the comment. We added one figure (figure 2) and one table (table 2) to give detailed reaction conditions and results. In addition, comparison of chemical and enzymatic methods was summarized and reviewed in revised manuscript.

Comment 2.  reaction conditions of chemical and enzymatic reactions should be compared (e.g., Temp., pH, solvent, time). a comprehensive table should be added that summarizes all enzyme-catalyzed N-amidation of nitrogen-containing heterocyclic reactions containing experimental conditions and yields.

Response to comment 2.  Thank you for the comment. We add a table to compare the advantages, disadvantages and reaction conditions of different methods.

Comment 3.  many related original and review papers are not cited

Response to comment 3.  Thank you for the comment. We renewed the reference list and added about 15 new references, correspondence part in the main text was also updated.

Round 2

Reviewer 2 Report

The manuscript was supplemented and it may be published in its present form.

Reviewer 3 Report

I believe the manuscript has been sufficiently improved to warrant publication in Bioengineering.